# COVID-19 Animal Models and Vaccines: Current Landscape and Future Prospects

**DOI:** 10.3390/vaccines9101082

**Published:** 2021-09-26

**Authors:** Shen Wang, Ling Li, Feihu Yan, Yuwei Gao, Songtao Yang, Xianzhu Xia

**Affiliations:** 1Key Laboratory of Jilin Province for Zoonosis Prevention and Control, Changchun Veterinary Research Institute, Chinese Academy of Agricultural Sciences, Changchun 130122, China; 18203762077@163.com (S.W.); xiaxzh@cae.cn (X.X.); 2National Research Center for Exotic Animal Diseases, China Animal Health and Epidemiology Center, Qingdao 266000, China; lling@cahec.cn

**Keywords:** COVID-19, SARS-CoV-2, animal model application, vaccine development

## Abstract

The worldwide pandemic of coronavirus disease 2019 (COVID-19) has become an unprecedented challenge to global public health. With the intensification of the COVID-19 epidemic, the development of vaccines and therapeutic drugs against the etiological agent severe acute respiratory syndrome coronavirus 2 (SARS-CoV-2) is also widespread. To prove the effectiveness and safety of these preventive vaccines and therapeutic drugs, available animal models that faithfully recapitulate clinical hallmarks of COVID-19 are urgently needed. Currently, animal models including mice, golden hamsters, ferrets, nonhuman primates, and other susceptible animals have been involved in the study of COVID-19. Moreover, 117 vaccine candidates have entered clinical trials after the primary evaluation in animal models, of which inactivated vaccines, subunit vaccines, virus-vectored vaccines, and messenger ribonucleic acid (mRNA) vaccines are promising vaccine candidates. In this review, we summarize the landscape of animal models for COVID-19 vaccine evaluation and advanced vaccines with an efficacy range from about 50% to more than 95%. In addition, we point out future directions for COVID-19 animal models and vaccine development, aiming at providing valuable information and accelerating the breakthroughs confronting SARS-CoV-2.

## 1. Introduction

In December 2019, a previously unknown beta coronavirus causing human pneumonia emerged and was soon isolated, named 2019-nCoV [1]. Subsequently, the virus was renamed SARS-CoV-2 and the syndrome was named COVID-19 by the World Health Organization (WHO) [2]. By 17th September 2021, over 226 billion COVID-19 cases have been confirmed, causing 4.65 million deaths worldwide [3]. The progression and dissemination of COVID-19 seriously threatened international health security and caused an immeasurable loss on the global economy. As a result, scientists all over the world are embarking on prophylactic and therapeutic research on SARS-CoV-2.

SARS-CoV-2 is an enveloped, positive-sense, single-stranded ribonucleic acid (RNA) virus encoding 16 non-structural proteins (nsp1-nsp16), several accessory proteins, and four structural proteins, including spike surface glycoprotein (S), matrix protein (M), small envelope protein (E), and nucleocapsid protein (N) [4]. S protein plays a key role in mediating the virus entry via interacting with the receptors of the host cell, and is considered as the main target to induce neutralizing antibodies (NAbs). S protein is composed of S1 and S2 subunits, of which the S1 subunit functions as a receptor-binding subunit while the S2 subunit mediates membrane fusion. Within S1, a region with 194 residues named the receptor binding domain (RBD) is identified as the core sequence for SARS-CoV-2 binding to a host cell. RBD shows a high binding affinity to human angiotensin-converting enzyme 2 (hACE2) and serves the entry and infection of SARS-CoV-2 [5,6]. Some enzymes, including transmembrane protease serine 2 (TMPRSS2), cathepsin B/L, and RNA-dependent RNA polymerase (RdRp) are key regulators of viral entry, replication, and transcription [5].

This life-threatening COVID-19 is characterized by symptoms of viral pneumonia, including fever, cough, and chest discomfort. In severe cases, dyspnea and bilateral lung infiltration are observed [1,7]. Generally speaking, elderly patients with comorbidities face a higher risk for SARS-CoV-2 infection and unfavorable prognosis. Current medical knowledge and research on COVID-19 for severe and critical patients’ management and experimental treatments are still evolving, but several protocols on minimizing the risk of infection among the general population, patients, and healthcare workers have been approved and diffused by International Health Authorities [8].

Besides the above issues about COVID-19, including clinical characteristics as well as countermeasures, animal models recapitulating the transmission characteristic, pathology, and corresponding immunological response to COVID-19 are foundational and urgent needs. Although great progress has been achieved in the development of prophylactic and therapeutic measures, only a few products have been proven effective. Currently, animal models involved in the study of COVID-19 include mice, golden hamsters, ferrets, nonhuman primates, and pigs. More than 302 vaccine candidates are under development, of which 117 of them have successfully entered clinical trials [9]. Herein, we briefly discuss popular animal models in vaccine evaluation and cutting-edge vaccines of COVID-19, followed by a detailed discussion of their commonalities and personalities as well as their pros and cons. We aim to grasp key information regarding animal models and vaccine research of COVID-19 and provide references for subsequent breakthroughs.

## 2. Animal Models for SARS-CoV-2 Vaccines Evaluation

Before clinical trials, COVID-19 vaccines need pre-clinical evaluations to ensure their safety and efficacy. COVID-19 animal models are fundamental and essential needs in this phase. Here, we retrospected the basic background and the development paths of COVID-19 animal models. More importantly, we paid close attention to the unique features that animal models required for COVID-19 vaccine evaluation. Animal models as well as susceptible animals (potential animal models) of COVID-19 were summarized in Figure 1. Characteristics of these animal models were summarized in Table 1.

### 2.1. Mouse Models

Mouse models are the most frequently used animal models in the preclinical study of vaccines. Mouse models are economical, abundant, well-characterized, easy to handle and manipulate, which are all characteristics that support the extensive development of mouse models. However, SARS-CoV-2 exhibited limited affinity to murine ACE2 [10], indicating that mouse models are less susceptible to SARS-CoV-2, which hindered the full application of mouse models. Subsequently, this issue was handled from three directions: hACE2 transgenic mice, hACE2 transduced mice as well as mice-adapted SARS-CoV-2.

In the hACE2 transgenic C3B6 mouse [11], a high viral load in lungs and pre-exposure protection were accomplished. Based on CRISPR/Cas9 knock-in technology, the mACE2 gene of the C57BL/6 mouse model was completely replaced with hACE2 (termed hACE2 mice) [12]. Viral loads, interstitial pneumonia, and elevated cytokines occurred in SARS-CoV-2 infected hACE2 mice. In hACE2 mice, the viral RNA load in the lungs was much higher and the distribution of hACE2 in various tissues was more in line with human conditions in comparison to other hACE2 genetically engineered mice generated by pronuclear microinjection [11,13]. In particular, the pathological changes observed in aged hACE2 mice were more obvious. Interestingly, intragastric infection of SARS-CoV-2 has been established in hACE2 mice, suggesting that the intestinal tract may be another transmission route of SARS-CoV-2. For mucosal-associated COVID-19 vaccines, especially oral vaccines, hACE2 mice are a reasonable choice. Besides, transgenic hACE2 mice models exhibited typical pathological changes in lungs for SARS-CoV-2-induced acute respiratory illness [14]. This is of significance for the evaluation of vaccines against SARS-CoV-2-induced severe acute respiratory distress syndrome (ARDS).

The hACE2-transduced models were established by transducing hACE2 mainly through replication-defective adenoviruses in BALB/c and C57BL/6 mice [15,16]. Corresponding clinical signs (weight loss, severe pulmonary pathology) and virus replication in the lungs were observed in hACE2-transduced models after SARS-CoV-2 infection. However, anti-vector immunity limits the full application of this animal model to a certain extent.

Mouse-adapted SARS-CoV-2 was obtained by serial passages, of which MASCp6 was achieved after six passages of SARS-CoV-2 in aged (9 months old) BALB/c mice. MASCp6 efficiently infected both the aged and young (6 weeks old) BALB/c mice. It replicated efficiently in the lung and trachea, resulting in moderate pneumonia as well as inflammatory responses [17]. A key substitution of N501Y in RBD was predicted to contribute to the enhanced infectivity of MACSp6 in mice. Besides, a mouse-adapted SARS-CoV-2 HRB26M efficiently infected the upper and lower respiratory tract of young BALB/c mice and C57BL/6J mice [18]. Subsequently, a lethal mouse-adapted SARS-CoV-2 MA10, which caused acute lung injury (ALI) in young and aged BALB/c mice, was isolated after ten passages in young BALB/c mice. It exhibited the epidemiological characteristics of COVID-19 disease as well as aspects of host genetics, age, cellular tropisms, elevated Th1 cytokines, and loss of surfactant expression and pulmonary function linked to the pathological features of ALI [19]. Interestingly, SARS-CoV-2 MA10 showed no mortality in ten-week-old C57BL/6J mice. The process of adaptation introduced multiple point mutations into the virus genome that are responsible for increasing virulence; yet whether this artificially-introduced genetic divergence compromises the relevance of the adapted viruses in the first place remains to be fully elucidated. As we expected, SARS-CoV-2 adapted in mice with a stronger physique or a younger age were more virulent. Lethal mouse-adapted SARS-CoV-2 has achieved breakthrough, with 100% fatality and clear mutation site (unpublished data). All mouse-adapted SARS-CoV-2 cause more severe disease in aged mice compared with young mice in the same manner.

### 2.2. Golden Hamster Models

Golden hamsters have previously shown susceptibility to severe acute respiratory syndrome coronavirus (SARS-CoV) [20]. They were another potential small animal model to study the pathogenesis and transmission of SARS-CoV-2 [21]. Weight loss, rapid breathing, and viral replication in the respiratory tract were observed in SARS-CoV-2 inoculated hamsters. They efficiently transmit SARS-CoV-2 to naïve hamsters by direct contact and via aerosols. SARS-CoV-2 infection in golden Syrian hamsters resembled features of mild COVID-19 patients [22]. Immunoprophylaxis hamsters with early convalescent serum significantly decreased the viral load in the lungs, but did not prevent or ameliorate lung pathology. SARS-CoV-2 infection triggers bronchopneumonia and a strong inflammatory response in the lungs with neutrophil infiltration and edema [23]. Several vaccine candidates have been tested in hamsters. Vaccinated hamsters showed a reduced or disappeared infectious virus in the lungs and less body weight loss, as well as less histopathological changes of pneumonia. They produced a level of NAbs against SARS-CoV-2 [24,25,26,27]. In short, golden hamsters reflect certain clinical manifestations and immune responses after the administration of COVID-19 vaccines.

### 2.3. Ferret Models

The domestic ferret (*Mustela putorius furo*) is naturally susceptible to many viruses including bunyaviruses, paramyxoviruses, rhabdoviruses, and togaviruses. Besides, ferrets have been used as a model system for respiratory diseases. They are naturally susceptible to the influenza A virus and recapitulate many aspects of an influenza infection. In addition, ferrets have been applied in COVID-19-related research [28]. Ferrets recapitulated some typical disease features of COVID-19 observed in humans. Naturally, infected ferrets rapidly transmitted SARS-CoV-2 to the entire population via direct or indirect contact [29]. Infected ferrets exhibited elevated body temperatures and virus replication. Virus shedding was confirmed in nasal washes, saliva, urine, and feces, while infectious viruses were detected in the nasal turbinate, trachea, lungs, and intestine, with acute bronchiolitis present in the infected lungs. Moreover, ferrets have been involved in the research field of viral behavior and host response [30]. SARS-CoV-2-post-infected ferrets revealed a unique and inappropriate inflammatory transcriptional response, which was characterized by low levels of type I and III interferons juxtaposed to elevated chemokines and a high expression of IL-6. Once again, it was proved in ferrets that the imbalanced host response to SARS-CoV-2 drove COVID-19.

Thus, ferrets represented an ideal animal model for virus shedding, transmission, and post-infection transcriptional response. However, mild clinical symptoms and relatively lower virus titers in lungs hindered the full application of ferret models. Due to the moderate susceptibility to SARS-CoV-2, ferrets were considered to have a mild clinical disease model of COVID-19. As shown in a previous study [31], Viral RNA shedding in the upper respiratory tract (URT) was observed in all ferrets (6/6) after a high (5 × 10^6^ plaque forming unit, PFU) dose of SARS-CoV-2 challenge, while only 1/6 ferrets showed similar signs after a low dose (5 × 10^2^ PFU) challenge. According to the above-discussed dose-dependent response to the infection of SARS-CoV-2, the application of ferrets in vaccine evaluation is relatively limited.

### 2.4. Nonhuman Primate Models

Due to the similar biological characteristics of non-human primates (NHPs) and humans, NHPs are seen as golden standard models of many emerging infectious diseases, including Ebola, Lassa fever, etc. [32,33,34,35,36,37,38,39].

In rhesus macaques models, modestly decreased appetite, changes in respiratory patterns, piloerection, and responsiveness were observed. Infectious SARS-CoV-2 were detectable mainly in the URT. Viral replication in aged (15 years) animals were more active than those of young (3–5 years) animals after SARS-CoV-2 challenge [40]. Infected animals developed typical interstitial pneumonia, especially in aged animals. They exhibited diffuse severe interstitial pneumonia. The respiratory disease caused by SARS-CoV-2 lasted 1–2 weeks in infected rhesus macaques [41]. Infiltration could be seen in lung radiographs of all infected animals. Mild transient neutropenia and lymphopenia were observed in the high dose group. Obvious clinical symptoms were not observed [42]. After infection, virus-specific antibody and NAbs responses started to appear at ten days post-immunization (dpi). Animals with the lowest and tardiest NAbs response showed prolonged viral shedding from the intestinal tract, which suggested that NAbs play a role in the control of infection. SARS-CoV-2 challenge and rechallenge trials in rhesus macaque models suggesting that all exposed macaques developed binding antibody response and NAbs to SARS-CoV-2 [42,43], while a large part of the animals developed cellular immunity. SARS-CoV-2 rechallenging animals showed reductions in median viral loads compared with their primary infection. Challenged rhesus monkeys were almost completely protected from the reinfection of SARS-CoV-2, indicating the production of protective immunity. Overall, rhesus macaques successfully recapitulated many hallmark features of human COVID-19 and triggered certain immune responses after SARS-CoV-2 infection. Inhibited virus replication and the production of certain humoral and cellular immunities against SARS-CoV-2 are two remarkable features exhibited in rhesus macaques after the vaccination of COVID-19 vaccines [44,45,46,47,48,49,50]. Antibodies decreased at seven months post-infection [44], which is similar to human beings. From this point of view, the duration of the immune response in rhesus monkeys can better reflect the actual situation in human beings.

Similarly, cynomolgus macaques were permissive to SARS-CoV-2 infection and displayed COVID-19-like disease [51]. SARS-CoV-2 replicated efficiently in respiratory epithelial cells throughout the respiratory tract. Prolonged viral shedding was observed in the URT of aged animals. All SARS-CoV-2 post-challenge remaining animals produced SARS-CoV-2–specific antibodies against the virus S1 domain and N proteins. In SARS-CoV infected cynomolgus macaques, lung lesions were typically more severe compared with SARS-CoV-2 infection. In the evaluation of COVID-19 vaccines, vaccinated cynomolgus macaques produced high levels of RBD-specific immunoglobulin G (IgG) and potent NAbs, and exhibited lower or no viral RNA copies and very mild histopathological changes in the lungs [52,53].

### 2.5. Other Susceptible Animals

In addition to the above commonly used laboratory animal models, the susceptibility of other animals to SARS-CoV-2 has been reported, including raccoon dogs [54], minks [55,56,57,58], dogs [59], jaguars, cats [60], fruit bats [61], tigers, gorillas, white-tailed deer [62], etc. In the Netherlands, United States, and Denmark, millions of minks have been culled over concerns that the animals transmit SARS-CoV-2 to human beings. Subsequently, cases of mink-to-human transmission were confirmed [55,56,57,58]. COVID-19 infects minks in a similar way to humans, causing respiratory symptoms and lung lesions that tend to be worse in aged animals. Besides, one-third of white-tailed deer in the north-eastern United States showed detectable antibodies against SARS-CoV-2, suggesting that wild animals may be widely exposed to SARS-CoV-2 [63]. In theory, animals susceptible to SARS-CoV-2 are potential animal models. On the other hand, the broad spectrum of SARS-CoV-2 infections is an overall concern. Attention should be paid to the prevention and control of SARS-CoV-2 infection in animals.

## 3. Vaccines for SARS-CoV-2 Prevention

We have summarized the most sophisticated COVID-19 vaccines, including inactivated vaccines, protein subunit vaccines, virus-vectored vaccines, and nucleic acid vaccines in Table 2.

### 3.1. Inactivated Vaccines

Using radiation techniques and chemical substances for viral inactivation, the inactivated vaccines are a member of the first generation vaccine that has been widely used for decades. The complete antigen epitopes of COVID-19 inactivated vaccines enables the full exposure of immune epitopes other than the S protein. However, the production of COVID-19 inactivated vaccines must be handled in the biosafety level 3 workshop, which poses the primary challenge. At present, three inactivated vaccines have been approved in China, whilst an inactivated vaccine named COVAXIN^®^ has been approved in India [65]. The research of inactivated vaccines in other countries is rarely reported.

Sinovac Biotech Ltd. (Beijing, China) developed an inactivated SARS-CoV-2 vaccine named CoronaVac. In preclinical trials, CoronaVac was tested in mice, rats, and NHPs [66]. Experiments on BALB/c mice and Wistar rats provided the primary information about safety and immunogenicity whilst defining the appropriate dose. Further evaluation in rhesus macaques confirmed the safety and immunogenicity data. Vaccinated macaques were protected from the SARS-CoV-2 challenge without the antibody-dependent enhancement (ADE) effect. In phase I/II of the clinical trials, within which the elderly were included, CoronaVac was safe and immunogenic [67]. Phase III clinical trials in Brazil showed that the protective efficacy of CoronaVac against COVID-19 at 14 dpi, the protection efficacy of cases requiring medical attention, and the protective efficacy of hospitalized cases were 50.65%, 83.70%, and 100.00%, respectively. In Turkey, the protective efficacy of CoronaVac 14 dpi against COVID-19 was 91.25% [68]. Sinovac Biotech Ltd. has established two production lines, which ensured the mass production of vaccines. On 5 February 2021, the conditional marketing application for CoronaVac was approved [69].

Developed by the Chinese Center for Disease Control and Prevention and Beijing Institute of Biological Products Company Limited, BBIBP-CorV is another licensed inactivated vaccine. Similarly, evaluation in mice, rats, guinea pigs, and rabbits provided initial immunogenicity data and optimized immunization doses and programs [70]. A challenge study was conducted in rhesus macaques to evaluate the protective efficacy. Significantly reduced SARS-CoV-2 titers and obvious NAbs were detected in rhesus macaques as compared to that of the placebo group. In the phase I/II clinical trials [71], BBIBP-CorV was safe and well-tolerated. The serum positive conversion rate reached 100% at 42 dpi. At this stage, the appropriate immunization procedure is preliminarily determined. Phase III trials showed that the protective efficacy of BBIBP-CorV against COVID-19 and the protective efficacy of hospitalized cases were 78.1% and 78.7%, respectively. In obese and elderly patients, the protective efficacy of BBIBP-CorV reached 80.7% and 91% (unpublished data). On 31 December 2020, BBIBP-CorV became the first inactivated vaccine approved for human use in China.

Sinopharm Wuhan Institute of Biological Products developed another inactivated vaccine, termed COVILO. In the phase I/II clinical trials [72], COVILO exhibited good safety and immunogenicity. On 28 February 2021, COVILO became the third COVID-19 vaccine licensed in China. In Phase III clinical trials, the protective efficacy of COVILO was 72.8% [73].

### 3.2. Protein Subunit Vaccines

Subunit vaccines are composed of immunogenic proteins or peptides of specific antigens. Characteristics such as safety, immunogenic, and flexibility shed light on the development of protein subunit vaccines [74], of which the S protein of SARS-CoV-2 with stabilized dimeric or trimeric form (S-Trimer) was a very active line, which overcame the limited immunogenicity of monomeric RBD [75,76,77].

The dimeric form of RBD was formerly used in Middle East Respiratory Syndrome coronavirus (MERS-CoV) vaccines [78]. When RBD-dimer was applied to SARS-CoV-2 and SARS-CoV vaccine development, 10–100 folds of enhancement of NAbs was achieved. Of which ZF2001, a tandem-repeat dimeric RBD protein-based COVID-19 vaccine [79], has been authorized by Uzbekistan for emergency use. ZF2001 appeared to be well-tolerated and immunogenic in phase I and II trials [80]. Moreover, a 25 μg dose in a three-dose schedule was finally determined for large-scale phase III trial evaluation.

Clover Biopharmaceuticals developed an S-Trimer subunit vaccine named SCB-2019 [81]. SCB-2019, combined with AS03 or CpG/Alum adjuvants, accomplished promising immunogenicity. However, in the phase I clinical trial, two serious adverse events were recorded, both in aged adults, including cellulitis and hyponatremia. Developed by Novavax, NVX-CoV is another distinguished S-Trimer-based nanoparticle vaccine [82]. In preclinical trials [83], hACE2-transduced BALB/c mice were applied in a challenge study, while baboons were used to supplement the immunogenicity data. The protective efficacy of the vaccine is reflected by a significant reduction in virus titers, protection against weight loss, and significantly reduced lung pathology and inflammation in mice. Preclinical studies optimized the optimal inoculation dose and highlighted the importance of adjuvants. The immunogenicity results of mice and baboons highly reflect the actual situation in phase I/II human clinical trials, especially the antigen dose-sparing effect of adjuvant [82]. Finally, a two-dose 5-μg adjuvanted regimen was determined.

### 3.3. Virus-Vectored Vaccines

Viral-vectored vaccines have been widely used for the prevention of emerging infectious diseases [84]. They are immunogenic due to their ability to enhance both humoral and cellular immune responses. Nevertheless, safety issues and pre-existing immunity against the vectors are the major concerns for viral-vectored vaccines [85]. At present, adenovirus vector, vesicular stomatitis virus, attenuated rabies virus, and modified vaccinia virus are the most popular virus vectors [86]. Herein, we mainly introduced the advances of the above viral-vectored vaccines of COVID-19.

#### 3.3.1. Adenovirus Vector

Adenovirus type 5 (Ad5) has several unique advantages as a vaccine vector, including prominent immunogenicity, easy manipulation, and a strong ability to express exogenous genes and to elicit a humoral and cellular immune response. Ad5-EBOV is one of the most advanced Ebola vaccines [87,88,89,90]. The Ad5-nCoV vaccine was developed by CanSino Biological Inc. and Beijing Institute of Biotechnology, which was designed to deliver the gene of the SARS-CoV-2 S protein into human cells. Ad5-nCoV is the first single-dose vaccine candidate to enter clinical trials. In the phase I clinical trial [91], Ad5-nCoV was tolerable in healthy adults. Adverse reactions were mild or moderate, including fever, fatigue, headache, and muscle pain, and no serious adverse event was noted. S-specific antibodies and NAbs increased significantly at 14 dpi, and peaked at 28 dpi. SARS-CoV-2 specific T cell response peaked at 14 dpi. However, high pre-existing Ad5 NAbs compromised the seroconversion of SARS-CoV-2 NAbs and reduced the peak of post-vaccination T cell responses. SARS-CoV-2 NAbs titers were relatively low, ranging from 14.5 to 34 at four weeks post-infection. In the phase II clinical trial [92], safety and immunogenicity were further assessed, especially in participants over 55. NAbs were consistent with the results in the phase I clinical trial. The results of the phase III clinical trial suggested that 14 or 28 days after a single dose injection of the vaccine, the overall protective efficacy was 68.83% and 65.28%, respectively. The protective efficacy against the occurrence of severe illnesses at 14 or 28 dpi was 95.47% and 90.07%, respectively [93]. On 5 February 2021, the conditional marketing application for Ad5-nCoV was approved.

Ad26-vectored vaccines have also been evaluated in clinical trials. Among them, Ad26-Ebola was confirmed to be a safe vaccine for humans with good immunogenicity [94]. Harvard Medical School constructed an Ad26-vectored vaccine expressing SARS-CoV-2 variants S with different leader sequences, antigen forms, and stabilization mutations [95]. According to safety and immunogenicity data, the NHPs experiment demonstrated the optimal vaccine candidate—termed Ad26.COV2.S—which contained the wildtype leader sequence, the full-length membrane-bound S with a mutation in the furin cleavage site, and two proline stabilizing mutations. A single shot of Ad26.COV2.S induced robust NAbs and provided complete or near-complete protection in bronchoalveolar lavage and nasal swabs following the SARS-CoV-2 challenge. Ad26.COV2.S is currently being evaluated in phase III clinical trials.

Replication-deficient chimpanzee adenovirus type 1 (ChAdOx1) has been utilized for MERS vaccine development [96,97]. The University of Oxford developed a ChAdOx1-vectored SARS-CoV-2 vaccine encoding a codon-optimized full-length S gene [98]. A single dose of ChAdOx1-S induced an immune response and reduced viral loads in rhesus macaques, but virus titers remained high in the upper and lower respiratory tract after the challenge. Three monkeys showed accelerated breathing and obvious symptoms after challenge. There was no difference in the viral load in the nasal swabs when compared with the control. Although no pneumonia or immune-enhanced diseases were observed, the efficacy of the vaccine was still questionable. In another study, ChAdOx1-S was tested in mice (BALB/c and CD1) and pigs [99]. A prime-boost strategy significantly enhanced antibody and T cell responses in pigs, but not in mice, compared to the single dose group. This study indicated that immunogenicity data in mice were distributed at the upper end of the dose response curve, which may saturate the immune response and largely obscure the difference between different regimens. Compared with mice, pigs may be a better animal model. In phase the I/II and II/III clinical trials [100,101], ChAdOx1 nCoV-19 was tolerable, and humoral and cellular immune responses were observed in a large part of volunteers. Homologous boosting increased antibody responses and protective efficacy, especially in those with a longer prime-boost interval (≥12 weeks). There were no hospital admissions for COVID-19 in the vaccination group after the initial 21-day exclusion period. The antibody and protection efficacy lasted at least 3-months. The UK regulatory authority has approved AZD1222 for emergency use. For participants who received two standard doses and participants who received a low dose followed by a standard dose, the reported efficacy was 62.1% and 90.0%, respectively, resulting in an overall efficacy of 70.4% [102]. However, research in South Africa indicated that the efficacy of ChAdOx1 nCoV-19 has been affected by the variant 501Y.V2, wherein it failed to resist the occurrence of mild or moderate COVID-19 in some cases. Moreover, on 10 September 2020, the Phase III clinical trial was once called off due to a case report of transverse myelitis in one volunteer. Subsequently, the European Drug Administration (EMA), the British drug regulatory agency (MHRA), and WHO issued statements that ChAdOx1 nCoV-19 may be associated with rare thrombosis, so the safety of ChAdOx1 needs further evaluation.

The heterologous prime-boost strategy is an effective scheme to reduce the pre-existing adenovirus immunity. Russia has developed a rAd26 and rAd5 vector-based heterologous prime-boost strategy, termed Sputnik V [103]. Compared with the single dose strategy, the heterologous rAd26 and rAd5 vector-based COVID-19 vaccine induced stronger humoral and cellular immune responses in participants. In Phase 1/2 studies, Sputnik V induced strong humoral responses in all participants, with a 100% seroconversion. In the Phase 3 trial, which involved almost 20,000 subjects, a 91.6% efficacy was reported [104].

#### 3.3.2. Vesicular Stomatitis Virus Vector

Rapid replication, high growth titer, multi antigens expression ability, and single dose use; these characteristics rendered the vesicular stomatitis virus (VSV)-vectored vaccine very popular in recent years. One of the most advanced Ebola vaccines, VSV-EBOV, was developed based on this platform expressing the glycoprotein of Ebola virus, and achieved promising results in clinical trials [105]. Case et al. constructed a replication-competent VSV-vectored vaccine that expressed a modified form of the SARS-CoV-2 S gene in place of the native glycoprotein gene (VSV-eGFP-SARS-CoV-2). Vaccinated mice produced a high level of NAbs and showed significantly reduced SARS-CoV-2 infection and inflammation in the lungs. A passive transfer of immunization sera conferred protection for naïve mice from the SARS-CoV-2 challenge. In addition to acting as a vaccine vector, VSV has been successfully applied in neutralization assay at biosafety level 2 and the mechanism study of SARS-CoV-2 infection [106,107]. Unfortunately, the clinical trials of the VSV-vectored COVID-19 vaccine progressed slowly due to safety issues.

#### 3.3.3. Rabies Virus Vector

The rabies virus (RABV)-vectored vaccine is usually used in the form of inactivation. Some progress has been achieved in the research of other coronaviruses, including SARS-CoV and MERS-CoV [108,109,110,111]. Wirblich, C. et al. developed a RABV-vectored vaccine candidate, CORAVAX™, containing the SARS-CoV S1 domain fused to the C-terminus of the RABV G protein [108,112]. Both live and inactivated candidates induced potent virus NAbs in BALB/c mice. According to previous experience in virus hemorrhagic fever, multi-dose inactivated vaccines based on rabies virus vectors were ideal in terms of the durability of immune protection [113,114,115].

### 3.4. Nucleic Acid Vaccines

Nucleic acid vaccines include Deoxyribonucleic acid (DNA) vaccines and RNA vaccines. As these are not virus-containing vaccines, a limited risk associated with virulence exists upon application. While this new technology brings surprises, there are many issues that are unknown. The strategy to improve in vivo transfection efficiency of nucleic acid vaccines is another challenge.

Inovio Pharmaceuticals was the first company to begin pre-clinical and clinical trials of a DNA vaccine (named INO-4800) against COVID-19. INO-4800 transfers DNA plasmids expressing the SARS-CoV-2 S proteins, which holds the preponderance of producing therapeutic antibodies and activating immune cells. The DNA vaccine was delivered to patients through the skin. INO-4800 induced T cell responses and NAbs responses against both the D614 and G614 SARS-CoV-2 S proteins [116]. Rhesus macaques vaccinated with two doses of INO-4800 were challenged 13 weeks after the second dose. During this period, the initial antibody and T cell immune response triggered by the vaccine had already declined. INO-4800 triggered a strong memory immune response to SARS-CoV-2. Although the immune response generated by memory B cells and T cells after challenge could clear the virus from the upper and lower respiratory tract more quickly, it could not completely prevent the infection and replication of the virus in the cells, indicating that this vaccine may help reduce the symptoms of COVID-19 in patients, but it failed to completely prevent the spread of the SARS-CoV-2.

Yu et al. developed several prototype DNA vaccine candidates expressing different forms of the SARS-CoV-2 S protein and evaluated their immunogenicity and protective efficacy in rhesus macaques [117]. Vaccinated macaques developed humoral and cellular immune responses. NAbs titers in the vaccinated macaques were comparable in magnitude to that of convalescent macaques and humans. A trend toward higher ADCD (antibody-dependent complement deposition) responses were observed in the S and S.dCT (deletion of the cytoplasmic tail) groups, while higher natural killer (NK) cell activation was observed in the RBD (receptor-binding domain with a foldon trimerization tag) and S.dTM.PP (a perfusion stabilized soluble ectodomain with deletion of the furin cleavage site, two proline mutations, and a foldon trimerization tag) group. SARS-CoV-2 challenged animals who were vaccinated with the vaccine encoding the full-length S protein resulted in reductions in median viral loads and increased cellular and humoral responses compared with sham controls. Vaccine-elicited NAbs titers correlated with protective efficacy. However, mild symptoms and a low level of virus replication were still observed in vaccinated macaques, which indicated that sterile immunity is not achieved. In the near future, the protective efficacy of the DNA vaccine needs to be further improved.

The mRNA molecule of the mRNA vaccine is rigorously modified and delivered via lipid nanoparticle systems. Based on the capacity of the individual to translate the encoding mRNA to specific antigens, the mRNA vaccine exhibits high expression efficiency and economical, cell-free, and scalable production capabilities. It is a hopeful alternative to traditional vaccines [118]. However, the mRNA vaccine shows poor stability, difficulty in administration, and relatively high probability of safety issues due to the unclear interaction of mRNA with the human body. Pfizer, Moderna, BioNTech, CureVac, Arcturus, and many other biological companies have established different types of mRNA platforms.

mRNA-1273 was developed by Moderna in collaboration with the US National Institute of Allergy and Infectious Diseases. Encapsulated by lipid nanoparticles, mRNA-1273 encodes a perfusion-stabilized viral S protein of SARS-CoV-2. mRNA-1273 induced robust SARS-CoV-2 NAbs in rhesus macaques, and achieved rapid protection in the upper and lower airways [119]. In contrast to the ChAdOX and DNA vaccine [117,120], evidence was provided that mRNA-1273 reduces viral replication in nasal tissue, eliminating the concern that other vaccines may not prevent the spread of the virus. In a dose-escalation phase I clinical trial, virus-specific antibodies increased in a dose-dependent manner, with the highest geometric mean titer (GMT) at 213, with 526 observed in the 250-μg group. NAbs were comparable to those of higher titers in convalescent patients. Systemic and severe adverse events were more likely to happen after the second vaccination and in the high dose group. In phase I/II clinical trials, 119 days after the first vaccination (two doses, 100 μg per dose, 28 days apart), binding antibodies and NAbs remained elevated in all participants 3 months after the booster vaccination. In the phase III trial, mRNA-1273 showed a 94.1% protective rate in an interim analysis [121,122].

Pfizer, in concert with BioNTech, developed other mRNA platform-based vaccines, including: BNT162b1, which encodes a secreted trimerized SARS-CoV-2 RBD; or BNT162b2, which encodes a membrane-anchored SARS-CoV-2 full length spike, stabilized in the prefusion conformation [123,124,125]. In the phase I clinical trial, BNT162b2 was safer than BNT162b1. Dose-dependent SARS-CoV-2 NAbs geometric mean titers were observed in both BNT162b1 and BNT162b2, with GMT (geometric mean titer) comparable or higher than that of SARS-CoV-2 convalescent serum samples. Considering several factors including safety and immunogenicity in the two phase I/II trials and NHPs challenge studies, BNT162b2 was finally selected for follow-up study. Despite the fact that it was lower than the NAbs titers against USA-WA1/2020 strain, BNT162b2 induced NAbs against engineered spike glycoproteins of emerged Delta variants at GMT of more than 40 [126]. BNT162b2 was reported to have a 95% protective efficacy in the phase III clinical trial involving 43,548 participants [124]. After two doses of vaccinations, protective efficacy ranged from 90% to 100% across subgroups defined by age, sex, race, etc. No differences existed between vaccine and placebo groups in terms of serious adverse events.

Qin et al. developed a lipid nanoparticle-encapsulated mRNA (mRNA-LNP) encoding a fragment of the RBD of SARS-CoV-2 (termed ARCoV) [127]. ARCoV induced NAbs as well as Th1-biased cellular response in BALB/c mice and cynomolgus monkeys. Two doses of ARCoV vaccinated in mice accomplished protection against the viral replication in the lower respiratory tract and lung lesions after the challenge of MASCp6.

## 4. Future Prospects of COVID-19 Animal Models and Vaccines

The year of 2021 will be an extraordinary year for the development and full deployment of COVID-19 vaccines. We summarized future prospects of COVID-19 animal models for vaccines from three aspects as well as COVID-19 vaccines from four aspects.

### 4.1. Prospects of Animal Models

#### 4.1.1. Qualified Animal Models

According to the animal rule issued by the FDA [131], when human efficacy studies are not ethical or feasible, evidence is needed to demonstrate the effectiveness of new drugs based on well-controlled animal studies, since the results of those studies establish that the product is reasonably likely to produce clinical benefit in humans. More precisely, there are four criteria for animal studies: a comprehensive reflection about pathophysiological mechanism of the toxicity; the effect should be demonstrated in more than one animal species to thoroughly predict the response in humans; the animal study endpoint should point to the desired benefit in humans; and the kinetics and pharmacodynamics in animals should allow preliminarily selection of an effective dose in humans.

For COVID-19, qualified animal models are required to have some common characteristics, including susceptibility to SARS-CoV-2 and the similar SARS-CoV-2 post-infection clinical manifestation as human beings. For vaccine evaluation, animal models should accurately reproduce the possible adverse reactions, the immune response, and the protective protection against SARS-CoV-2 challenge in human beings after vaccination. In the near future, protective correlation in more animal models will be elucidated to accurately reflect the immunogenicity of vaccines. Moreover, a challenge study in animal models should be compared with human phase III clinical trials, and animal models that directly reflect the protective efficacy of COVID-19 vaccines should be validated.

#### 4.1.2. Animal Model Selection Strategy for Vaccines

To obtain comprehensive preclinical information, two animals models are usually demanded. One provides the primary data about the safety and immunogenicity, while the other provides detailed data about the protective efficacy. A virus challenge should be conducted to support the protective efficacy of the vaccine if the condition permits. The former animal model should be economic, easily accessible, and operative, while the latter should be able to recapitulate the immune response in human beings. After evaluation in animal models, issues about safety, immunogenicity, protective efficacy, the immune dose, and the immune procedure should be preliminarily determined. According to the above discussion, mice-adapted SARS-CoV-2 or rhesus macaques will be an indispensable choice for two reasons. First, a protective correlation has been identified in mouse models, with a cut-off value of ~1:1009 PRNT 50 as full protection of SARS-CoV-2 lung infection [127]. In addition, the protective immunity of SARS-CoV-2 has been consistently elucidated in mice and rhesus macaques, where NAbs play a role in the control of infection [42,97,119,129]. To sum up, mice-adapted SARS-CoV-2 provides an economic, productive, and available small animal model for virus challenge, while rhesus macaques are the most comprehensively applied NHPs that accurately reflect the post-vaccinated immune response in human beings.

#### 4.1.3. Animal Model for Special Issues

No animal model recapitulates all aspects of human COVID-19. Beyond the above-mentioned common characteristics, the choice of animal models is determined by the purpose of the study. For instance, when we are committed to solving the problem that viruses still exists in the URT after vaccine immunization and eliminate the potential risk of transmission, golden hamster is the best choice for their obvious viral load of the URT after challenge and for their excellent performance in SARS-CoV-2 transmission experiments [22]. For SARS-CoV-2-induced ARDS, as well as a preliminary exploration of oral COVID-19 vaccine, hACE2 transgenic mice will be the best choice. For asymptomatic carriers or mild COVID-19 related research, ferrets and golden hamsters are both competent. Therefore, in subsequent animal model development, the detailed characteristics of each animal model will be fully displayed; that is, a precise definition of each animal model will be outlined for its application scope.

### 4.2. Prospects of Vaccines

#### 4.2.1. Safety, Immunogenicity, and Durability

Safety is the primary concern in vaccine development. of which immunopotentiation is the core, especially the ADE effect [132,133,134]. According to the vaccine development progress of SARS-CoV, vaccines of varying design strategies are all possible to cause ADE [135,136]. In May 2021, ADE regarding SARS-CoV-2 was clarified for the first time [137]. N-terminal-domain (NTD) targeted monoclonal antibodies who recognize specific sites were screened; they enhanced the binding capacity of the spike protein to ACE2 and the infectivity of SARS-CoV-2. The above antibodies were detected at high levels in severe patients, and were also found in uninfected donors. A key measure to prevent ADE is to select the appropriate target antigen and reduce the non-NAbs induction area, which means that a delicate balance between the size and the immunogenicity of the antigen should be achieved. High titers of NAbs and moderate cellular immunity increase the chance of treatment and reduce the risk of ADE, while antigens that induce large amounts of Th2 cytokines (such as IL-5 and IL-13) are prone to cause ADE [138,139].

When it comes to immunogenicity, the elucidation of protective immunity is significant. NAbs have been proven to play a role in the control of SARS-CoV-2 infection [42,97,119,129]. Innate immune effector functions such as ADCD may provide support [117]. Beyond humoral immunity, broad and strong memory CD4+ and CD8+ T cells are detected in convalescent COVID-19 patients [140]. However, it is uncertain what the role of virus-specific T cells is in the control and resolution of SARS-CoV-2 infections. In severe COVID-19 patients with ARDS, the strongest T cell responses were directed to S protein, and SARS-CoV-2-specific T cells predominantly produced effector and Th1 cytokines [141]. SARS-CoV-2-specific T cells are present relatively early and increase over time. The potential variations in T cell responses may serve as a function of disease severity and an indicator to understand the potential role of immunopathology in the disease. In current opinion, robust humoral and moderate cellular immunogenicity are both needed to increase the likelihood of inducing protection.

Since the first vaccine entered in clinical trials has only been tested for about one year, data about the durability is very limited. At least for now, it is certain that the vast majority of infected individuals with mild-to-moderate COVID-19 experience robust IgG antibody responses and NAbs persist 6–8 months after infection [142,143].

#### 4.2.2. Develop Vaccine from Multiple Design Strategies

For emerging infectious diseases, preexisting sophisticated technology provides a timely response. Several inactivated vaccines developed in China are staying in the frontier of the vaccine echelon. Despite the existence of uncertainty, emerging technology like mRNA and S trimer vaccines brings us surprising immune responses and up to 95% efficacy. They may be the best choice in terms of protective efficacy. From this point of view, the multi-line development of COVID-19 vaccine candidates enables specific people with specific choices. For frontline healthcare workers with corresponding medical conditions, two doses of an inactive vaccine or an mRNA vaccine would be a better choice. For underdeveloped regions like West Africa with poor medical conditions, maybe a single dose of an Ad5-S vaccine is more feasible. More importantly, the multi-line development of COVID-19 vaccines is the foundation of sequential immunization. Sequential immunization is an immune strategy that complements the advantages of different COVID-19 vaccines. German scientists tried a heterologous ChAdOx1 nCoV-19 and BNT162b2 prime-boost sequential immunization strategy. Compared with two doses of the BNT162b2 vaccination, the neutralization activity of SARS-CoV-2 variant B.1.1.7 was increased by four times [144]. In the near future, research related to sequential immunity will be widespread.

#### 4.2.3. Vaccine Based on the Mucosal Immune Pathway

Currently, most of the COVID-19 vaccines are delivered by intramuscular injection. However, COVID-19 vaccines delivered by intramuscular injection may not be able to inhibit the infection of the virus in the URT [145]. In addition, ACE2 is found to be highly concentrated in the oronasal epithelium [146], which explained the profound viral replication in the mucosal sites. In this situation, mucosal immunity is essential for blocking the viral entry through oro-respiratory tracts.

Although mucosal immune oriented COVID-19 vaccines lag behind, inspired by the oral polio vaccine, several nasal drops or oral COVID-19 vaccines have been reported. Vaxart has recently developed an enteric-coated tablet vaccine containing an adenoviral-vector that encodes the S and the N gene of the SARS-CoV-2 [147]. At present, there are no reports about the efficacy of the oral COVID-19 vaccine. As for the nasal drop vaccine, two independent research groups developed a Newcastle disease virus (NDV)-based COVID-19 vaccine, which expressed the S protein of SARS-CoV-2, termed NDV-S [148,149]. NDV-S elicits high levels of antibodies, and protects mice from a mouse-adapted SARS-CoV-2 challenge when the vaccine is given intramuscularly in mice. After two dose intranasal administrations of NDV-S, hamsters were completely protected. Importantly, virus shedding in nasal turbinate was significantly reduced. In addition, there are ideas of recombinant poliovirus Sabin, as well as respiratory viruses as delivery vectors [150,151]. Our team is doing similar work. For our technical route, we package our antigen through poly (lactic-co-glycolic acid) (PLGA), a Food and Drug Administration (FDA) approved cutting-edge drug adjuvant material. The formulation of PLGA with antigens and traditional Chinese medicine Dendrobium polysaccharides allows not only the delivery of antigens but also an adjuvant effect itself [152]. PLGA is modified by Polyethyleneimine, and linked with the M cell-targeted peptide (as described in Figure 2). The enteric coating prevents the contents’ active ingredient from the stomach’s acidic environment, so antigens reach the intestinal tract and are recognized by M cells. After ingestion, antigens are transferred to antigen-presenting cells to initiate an immune response.

#### 4.2.4. Battle with SARS-CoV-2 Variants

The convalescent COVID-19 patients re-infected with SARS-CoV-2 is a serious challenge. Among them, virus mutations are responsible for this phenomenon [153]. In March 2020, an increasingly prevalent SARS-CoV-2 variant encoding a D614G mutation in the viral S gene was reported. D614G is located in one of the predicted B-cell epitopes of the SARS-CoV-2 S protein, and this is a highly immunodominant region [154]. The S-G614 protein contains a novel serine protease cleavage site, so it could be cleaved by serine protease elastase-2 more efficiently, and thus entry efficiency is increased. This explains why it transmits more efficiently and is more effective at transducing cells [155,156,157,158]. G614 is more pathogenic, characterized by an increasing case fatality rate [159]. More recently, several new SARS-CoV-2 variants were reported, including the United Kingdom variant N501Y.V1 (B.1.1.7) [160], the South Africa variant N501Y.V2 (B.1.351) [161], the Brazil variant 501Y.V3 (P.1), the Indian variant Delta (B.1.617.2), the Lambda [162], as well as an animal variant cluster 5 in domestic minks. Several previous studies have confirmed that the above SARS-CoV-2 variants reduce the neutralizing activity of antibodies to different degrees [162,163,164,165,166,167,168]. Among them, the Delta variant is spreading all over the world: its spike P681R mutation accelerates and enhances S-mediated fusion [168], which may explain the shorter incubation period of Delta variant infection [169]. The control of the Delta variant is vital to curb the second wave of the epidemic. Although the mutation of SARS-CoV-2 has not yet had a disruptive impact on the effectiveness of the vaccine [126,166,170], preparations should be made right now to deal with SARS-CoV-2 variants. Close monitoring should be exercised on the protective efficacy of existing vaccines on SARS-CoV-2 variants. Once the vaccine loses its protective efficacy on emerging variants, mature technology routes should be adopted again. When new variants are added at the feeding end, COVID-19 inactivated vaccines confronting SARS-CoV-2 variants is produced without any changes in the production process. On the other hand, emerging vaccine design strategies should play a role in the rapid response of SARS-CoV-2 variants due to their flexibility. Sinovac, Moderna, and Pfizer have already initiated the boost dose of COVID-19 vaccine confronting the SARS-COV-2 variants. To sum up, mature and emerging flexible vaccine design technologies are powerful weapons against SARS-CoV-2 variants.

## 5. Conclusions

This review summarizes currently available animal models and promising vaccines, including their current stage, progress that has been achieved, and problems to be solved urgently. The core goal of this review is to grasp the universal connection of things and to integrate all aspects of SARS-CoV-2 vaccine development into a whole, providing useful and accessible information in hopes of accelerating the elimination of COVID-19 shortly. Once vaccination starts and herd immunity is established, the elimination of SARS-CoV-2 is in the near future.

## Figures and Tables

**Figure 1 vaccines-09-01082-f001:**
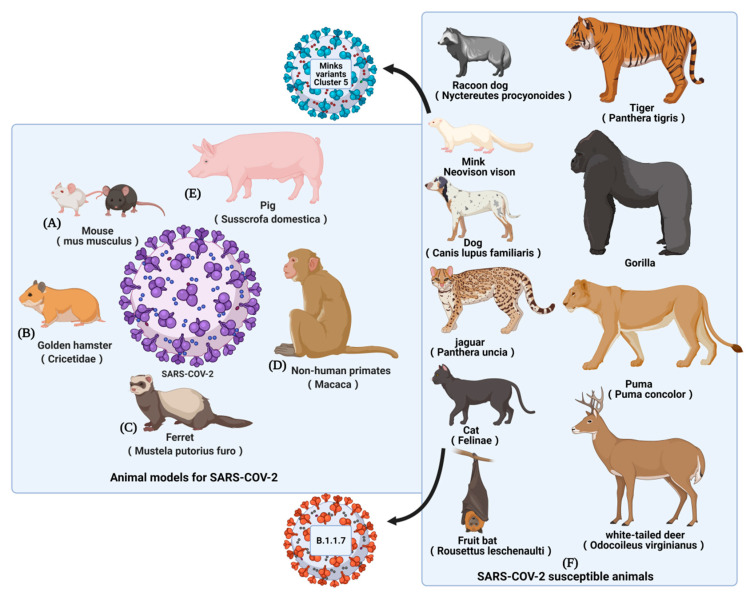
COVID-19 animal models and susceptible animals. (**A**) Mouse models: hACE2 transgenic mouse, hACE2-transdued mouse, and mouse-adapted SARS-CoV-2. (**B**) Golden hamster model. (**C**) Ferret model. (**D**) No human primates: Rhesus macaque model and Cynomolgus macaque model. (**E**) Pig model. (**F**) Other susceptible animals.

**Figure 2 vaccines-09-01082-f002:**
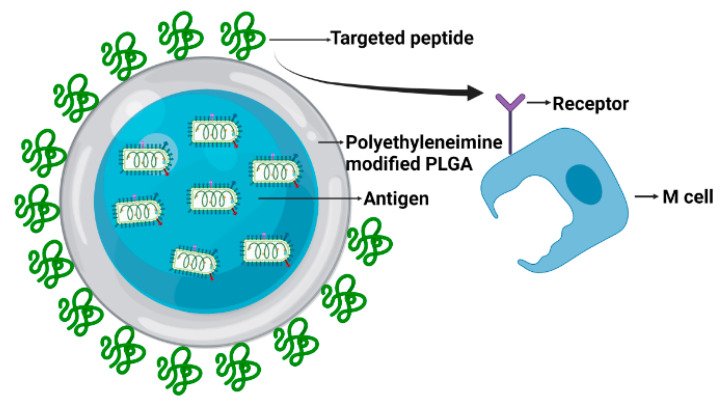
Schematic design of PLGA encapsulated oral COVID-19 vaccine: immunogenic antigen are packaged with PLGA in the form of water-oil-water, then modified with M cell receptor-targeted peptide, targeting M cells in the small intestine and stimulating a mucosal immune response.

**Table 1 vaccines-09-01082-t001:** Characterizations of animal models for COVID-19.

Animals/Design	Challenge Dose	Route	Lethality	Clinical Features	Infected Organs	References
hACE2 transgenic C3B6 mice	3 × 10^4^ TCID_50_	i.n.	No	Interstitial pneumonia and pathology, weight loss	Lungs, eye, heart, and brain	[11]
hACE2 transgenic C57BL/6 mice	4 × 10^5^ PFU 40 μL, 10^7^ PFU/mL	i.n./i.g. i.t.	No No	Interstitial pneumonia, pathology, and elevated cytokines ARDS, lung pathology, neutrophilic infiltration	Lung, trachea, and brain	[12,13] [14]
hACE2-transduced BALB/c and C57BL/6 mice	10^5^ PFU	i.n. + i.t.	No	Pneumonia, lung pathology, and weight loss	Lung, heart, spleen, and brain	[15,16]
BALB/c or C57BL/6 mouse-adapted SARS-CoV-2	7.2 × 10^5^ PFU 10^6.2^ PFU/10^4.4^ PFU 10^2^~10^5^ PFU	i.n. i.n. i.n.	No No Yes	Moderate pneumonia and inflammatory responses / ALI, lung disease, elevated cytokines	Lung, upper and lower respiratory tract	[17] [18] [19]
Golden hamsters	8 × 10^4^ TCID_50_/10^5^ PFU	i.n.	No	Lung pathology, weight loss, rapid breathing	URT, duodenum epithelial cells, and lung consolidation areas	[20,21,22,24,25,26,27,64]
Ferrets	10^5.5^ TCID_50_	i.n.	No	Elevated body temperature, acute bronchiolitis	Nasal turbinate, trachea, lungs, and intestine	[28,29]
Rhesus macaques	10^6^ TCID_50_	i.n.	No	Interstitial pneumonia and pathology, weight loss, asthenia, respiratory disease	Nose, throat, lung, and anus	[40,41,42,43,44,45,46,47,48,49,50]
Cynomolgus macaques	/	i.n. + i.t.	No	Lung pathology, no overt clinical signs	Nose, throat trachea, bronchi, and lung lobes	[51,52,53]

**Note:** TCID_50_: median tissue culture infective dose; i.n.: Intranasal inoculation; i.g.: Intragastric inoculation; i.t.: Intratracheal inoculation.

**Table 2 vaccines-09-01082-t002:** Details of vaccines for COVID-19.

Vaccine Design	Name	Country	Current Stage	Dose	NAbs (GMT)	Efficacy	Note	References
Inactivated vaccine	CoronaVac	China	Multinational EUA	2	23.8~44.1	50.65%~91.25%	Safe in the elderly and juveniles	[66,67]
	BBIBP-CorV	China	Multinational EUA	2	/	79.34%	Safe, pilot-scale production	[70]
	/ Covaxin	China India	Multinational EUA India EUA	2 2	121~247 /	72.51% 81%	Safe /	[72] [65]
Virus-vectored vaccine	ChAdOx1	Britain	Multinational EUA	2	274 (232~542)	66.7%	Reduced efficacy in the variants, adverse effects	[98,99,100,101,102]
	Convidicea	China	Multinational EUA	1	18.3~19.5	70.4%	Tolerable, safe in elder people, pre-existing Ad5 immunity	[91,92]
	Ad26-S Sputnik V	America Russia	America EUA Multinational EUA	1/2 2	113/600 44.5 (31.8–62.2)	66% 91.6%	Adverse effects Immunogenic in older	[95] [103,104]
	CORAVAX™	America	Phase I/II	1/3	/	/	Safe, long-lasting protection.	[112]
Nucleotide vaccine	INO-4800	America	Phase II	2	PNT:70~170	/	Antibody responses against both the D614 and G614 SARS-CoV-2	[117]
	bacTRL-S-1	America	Phase I/II	2	IC_50_:P: ~27 (W12)	/	Multiforms, reduce median viral loads	[116]
	mRNA-1273	America	Multinational EUA	2	PRNT_80_: 339.7; 654.3	94.5%	Antibodies remained more than 3 months	[121,122]
	BNT162b2	America	Multinational EUA	2	NT: 540; PNT: 10,000	95%	Antibody persisted for at least 70 days	[124,128]
	ARCoV	China	Phase I/II	2	NT_50_: ~1/699, ~1/6482	/	Completely protect mice against the challenge, thermostable	[127]
	saRNA LNP	Britain	Phase I/II	2	NT: 80 to 20,480	/	Highly immunogenetic	[129]
	Nanoparticle	America	Phase I/II	2	IC_50_: 3 × 10^3^ to 7 × 10^3^	/	Robust nAbs targeting distinct epitopes, stability, highly scalable	[130]
Subunit vaccine	SCB-2019	Australia	Phase III	2	1280~3948/1076~3320	/	Need adjuvant, robust immune responses	[81]
	NVX-CoV2373	America	Phase III	2	3906	89.33%	/	[82]
	ZF1001	China	Multinational EUA	2/3	102.5	/	Neutralizing 501Y.V2	[79,80]

**Note:** EUA: emergency use approval; IC_50_: the half maximal inhibitory concentration; NT: neutralizing titers; NT_50_: half maximum neutralization potency; PNT: pseudovirus neutralizing titers; PRNT_80_: plaque-reduction neutralization testing assay that shows reduction in SARS-CoV-2 infectivity by 80% or more. NAbs titers list from low to high dose.

## Data Availability

Not applicable.

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
