# Peer review of "COVID-19 Animal Models and Vaccines: Current Landscape and Future Prospects"

_vaccines, 2021, doi:10.3390/vaccines9101082_

Round 1
Reviewer 1 Report
The authors aimed to review briefly review popular animal models in vaccine evaluation and cutting-edge vaccines of COVID-19.
This an interesting review, well divided in the sub-paragraphs. that try to cover the key information of animal models and vaccine research of COVID-19 and provide reference for subsequent breakthroughs, but some issues should be improved.
The manuscript needs grammar correction.
Introduction section: will be very useful for the readers to stress better the concept as also stated from the authors that generally speaking, elderly patients with comorbidities face higher risk for SARS-CoV-2 infection and unfavorable prognosis (line 54-55).
I suggest to add the following paragraph:
Current medical knowledge and research on COVID-19 severe and critical patients' management and experimental treatments are still evolving, but several protocols on minimizing risk of infection among the general population, patients and healthcare workers have been approved and diffused by International Health Authorities (reference doi: 10.12998/wjcc.v8.i18.3920).
Author Response
Response to Reviewer 1 Comments
Point 1:
The manuscript needs grammar correction.
Response 1:
Thanks for your comments. We have thoroughly checked and corrected the grammar of our manuscript.
Point 2:
Introduction section: will be very useful for the readers to stress better the concept as also stated from the authors that generally speaking, elderly patients with comorbidities face higher risk for SARS-CoV-2 infection and unfavorable prognosis (line 54-55).
I suggest to add the following paragraph:
Current medical knowledge and research on COVID-19 severe and critical patients' management and experimental treatments are still evolving, but several protocols on minimizing risk of infection among the general population, patients and healthcare workers have been approved and diffused by International Health Authorities (reference doi: 10.12998/wjcc.v8.i18.3920).
Response 2:
Thanks very much. The paragraph that the reviewer suggested to add is essential to our manuscript. We have added it in line 55-61 and make it a part to better stress the concept.
Line 53-63:
Generally speaking, elderly patients with comorbidities face a higher risk for SARS-CoV-2 infection and unfavorable prognosis. Current medical knowledge and research on COVID-19 severe and critical patients' management and experimental treatments are still evolving, but several protocols on minimizing risk of infection among the general population, patients and healthcare workers have been approved and diffused by International Health Authorities [9].
Besides the above issues about COVID-19 including clinical characteristics as well as countermeasures, animal models recapitulating the transmission characteristic, pathology and corresponding immunological response to COVID-19 are foundational and urgent need.
Reviewer 2 Report
Xianzhu Xia and colleagues report on the status of animal models of COVID-19 and their utility for examining the immunogenicity and protective efficacy of COVID-19 vaccines. The review emphasizes the strenghts of the animal models, but briefly acknowledges the weaknesses such as lack of pre-existing immunity, co-morbidities, or "long haulers".
The manuscript requires a moderate level of proofreading and inclusion of some references. For the ferret model, Blanco-Melo (Cell 2020 PMID 32416070), et al., reported on the predictive value of the ferret model of COVID-19, which reflected some of the transcriptional responses seen in humans.
Section 4.1.1. "Qualified Animal Models" should include reference of the two-animal rule by the U.S. Food and Drug Administration. Within this section, the summary of NDV vaccines should include reference of W. Sun, et al., (EBioMedicine 2020 PMID 33232870), who developed a recombinant, replication competent NDV vectored spike-based vaccine.
Figure 1 includes common or scientific names for the depicted animals. For consistency, either the commor or scientific name should be provided for all species.
Author Response
Response to Reviewer 2 Comments
Point 1:
The manuscript requires a moderate level of proofreading and inclusion of some references.
Response 1:
Thanks for your comments. We have made an overall check of the references. These are revisions or updates of reference.
Pervious references 7 and 65 were removed.
Reference 8, 9, 23, 31, 32, 55, 100, 101, 127, 135, 144, 162, 164 and 165 was changed or updated.
Point 2:
For the ferret model, Blanco-Melo (Cell 2020 PMID 32416070), et al., reported on the predictive value of the ferret model of COVID-19, which reflected some of the transcriptional responses seen in humans.
Response 2:
We have added additional information about the ferret model as suggested.
Line 167-175:
Moreover, ferret has been involved in the research field of viral behavior and host response [32]. SARS-CoV-2-post infected ferrets revealed a unique and inappropriate inflammatory transcriptional response, which was characteristed by low levels of type I and III interferons juxtaposed to elevated chemokines and high expression of IL-6. Once again, it was proved in ferrets that the imbalanced host response to SARS-CoV-2 drove COVID-19.
Thus, ferret represented an ideal animal model for virus shedding, transmission and post-infection transcriptional responses seen in humans.
Point 3:
Section 4.1.1. "Qualified Animal Models" should include reference of the two-animal rule by the U.S. Food and Drug Administration.
Response 3:
We have included reference of animal rule by the U.S. Food and Drug Administration.
Line 513-519:
According to the animal rule issued by the FDA [135], when human efficacy studies are not ethical or feasible, evidence is needed to demonstrate the effectiveness of new drugs based on well-controlled animal studies, since the results of those studies establish that the product is reasonably likely to produce clinical benefit in humans. More precise, there are four criteria for animal studies: a comprehensive reflection about pathophysiological mechanism of the toxicity; effect should be demonstrated in more than one animal species to thoroughly predicting the response in humans; the animal study endpoint point to the desired benefit in humans; the kinetics and pharmacodynamics in animals allows preliminarily selection of an effective dose in humans.
Point 3:
Within this section, the summary of NDV vaccines should include reference of W. Sun, et al., (EBioMedicine 2020 PMID 33232870), who developed a recombinant, replication competent NDV vectored spike-based vaccine.
Response 3:
The reference of (EBioMedicine 2020 PMID 33232870) has been added in line 626. In line 622-629, we have rewrited the summary of NDV-vectored COVID-19 vaccine, covering the protective efficacy after intramuscular or intranasally delivery.
Line 622-629:
At present, there are no reports about the efficacy of oral COVID-19 vaccine. As for nasal drop vaccine, two independent research group developed a Newcastle disease virus (NDV)-based COVID-19 vaccine, which expressing the S protein of SARS-CoV-2, termed NDV-S [150,151]. NDV-S elicit high levels of antibodies and protect mice from a mouse-adapted SARS-CoV-2 challenge when the vaccine is given intramuscularly in mice. After two dose intranasal administrations of NDV-S, hamsters were completely protected. Importantly, virus shedding in nasal turbinate was significantly reduced.
Point 4:
Figure 1 includes common or scientific names for the depicted animals. For consistency, either the common or scientific name should be provided for all species.
Response 4:
For consistency, we provide common name and scientific name for all depicted animals animal models in figure 1. For gorilla, common name is same to scientific name.

Reviewer 3 Report
General comment: The authors presented an interesting review work concerning to the animal models available for COVID-19 vaccines development.The manuscript is well structured and well written.
Title: It clearly reflects the manuscript content.
Abstract: The abstract is adequately structured. The keywords should be different from those used in the title.
Figures 1 and 2. Are the images presented in this Figure original?
Recommendation: The manuscript should be accepted for publication in the present form.
Author Response
Response to Reviewer 3 Comments
Point 1:
The keywords should be different from those used in the title.
Response 1:
Thanks for your comments. Keywords have been changed to ‘COVID-19; SARS-CoV-2; animal model application; vaccine development’.
Point 2:
Figures 1 and 2. Are the images presented in this Figure original?
Response 2:
All figures are our original images and were designed by ourselves. In the revised version, we provide figures with higher resolution.
This manuscript is a resubmission of an earlier submission. The following is a list of the peer review reports and author responses from that submission.
Round 1
Reviewer 1 Report
Yan, Gao and co-workers summarize in the manuscript submitted to Vaccines the current state of available animal models for COVID-19. Most of the cited literature is correctly updated. Unfortunately, this referee feels that the reported data is not suitable for publication as a Review. Indeed, this is a very active line, which is currently ongoing. A large number of groups is developing such animal models, as illustrated in Table 2. Consequently, any review will be quickly out of date.
I am positive that such information better fits to other format, e.g., a contribution as Perspectives. Please, notice that Vaccines only accepts three types of publications: Articles, Review and Case reports. The submitted manuscript does not fit to any of such formats.
I strongly recommend resubmitting to a more suitable journal that allow for such Perspectives manuscript.
In addition, all figures must be rendered with a much larger resolution. The current state is quite poor.
Author Response
Response to Reviewer 1 Comments
Thank you very much for your thorough review of our manuscript. We have carefully addressed each of your comments below in a point-by-point manner.
Specific Comments:
Point 1: Unfortunately, this referee feels that the reported data is not suitable for publication as a Review. Indeed, this is a very active line, which is currently ongoing. A large number of groups is developing such animal models, as illustrated in Table 2. Consequently, any review will be quickly out of date.
Response 1: We agree with the reviewer to some extent, but we respectfully hold that our manuscript can be presented in the form of a review. We have made careful revisions to the structure of this manuscript to make it more in line with the criteria of the review. Secondly, COVID-19 animal models and vaccines are indeed developed at high speed, which posed a challenge to keep our manuscript up to date. To overcome this issue, we closely tracked and focused on cutting-edge COVID-19 vaccines and commonly used animal models rather than all related information. To sum up, our manuscript can reflect the background, research status and development trends of our topic, and it meets the stylistic requirements of the review.
Point 2: I am positive that such information better fits to other format, e.g., a contribution as Perspectives. Please, notice that Vaccines only accepts three types of publications: Articles, Review and Case reports. The submitted manuscript does not fit to any of such formats. I strongly recommend resubmitting to a more suitable journal that allow for such Perspectives manuscript.
Response 2: We respectfully disagree with the reviewer. Our manuscript did make some perspectives. More precisely, they are the prospects of future trends. These prospects are based on our review of the current landscape of COVID-19 animal models and vaccines. So we are positive that the core of our manuscript is to review progress that has been achieved in cutting edge COVID-19 animal models and vaccines, then provide valuable information and prospects. We would prefer it if we submitted our manuscripts to Vaccines. We sincerely believe that our review and prospects on the COVID-19 vaccine and animal models are in line with the requirements of this journal for review articles.
Point 3: In addition, all figures must be rendered with a much larger resolution. The current state is quite poor.
Response 3: Thanks for your comments. We have rendered all figures with a larger resolution according to the requirement this journal.
Thanks again for your hard work.
Reviewer 2 Report
I have carefully read the manuscript entitled "COVID-19 animal models and vaccines: current landscape and future prospects" which describes the current status, examples of past use, and future possibilities of using animal models in COVID-19 research.
It seems to be needed more detailed explanations before explaining the whole sentences.
For example, there is no information about other genetically engineered mice in details such as what genes were engineered in 93rd sentence and about what is the age of the mice from which the fatality rate was derived in 124th sentence. Also, I recommend writing more detail about what clinical symptoms are mild or moderate: ex) fever, injection area pain, etc in the 299th sentence.
In minor,
In table 1, there is a difference in quantity, the reliability of the sensitivity comparison by species is lowered.
In table 2, add NT50 and additional explanation is required for the asterisk.
In figure 2 and 3, writing more explanation is needed, we think.
In 4.5 title, please write more clearly about the prospects.
In minor, please re-check the number of units in many parts.
And there are some mis-spellings, non-grammatical sentences. It will seem to be a more smooth and meaningful sentence after corrections.
Please double-check the position of the exponent used in the table.
It seems that there are parts that need detail in word selection. For example, RNA and mRNA.
Author Response
Response to Reviewer 2 Comments
Thank you very much for your thorough review of our manuscript. We have carefully addressed each of your comments below in a point-by-point manner.
Point 1: It seems to be needed more detailed explanations before explaining the whole sentences. For example, there is no information about other genetically engineered mice in details such as what genes were engineered in 93rd sentence and about what is the age of the mice from which the fatality rate was derived in 124th sentence. Also, I recommend writing more detail about what clinical symptoms are mild or moderate: ex) fever, injection area pain, etc in the 299th sentence.
Response 1: Thank you very much for your suggestions. We have made the following revisions as suggested:
93rd (now 106th): “other genetically engineered mice” has been changed into “other hACE2 genetically engineered mice generated by pronuclear microinjection”.
124th (now 140th): 10 weeks old and 1 year old BALB/c mice
299th (now 336th): add: injection site pain was the most frequently observed, followed by fever.
Besides, we have added details in 209th, 396th,etc.
Point 2: In table 1, there is a difference in quantity, the reliability of the sensitivity comparison by species is lowered.
Response 2: Due to the availability of different species and different study designs, differences did exist in quantity of different species. We apologized that it is hard to perform head-to-head comparison between groups. To overcome this issue, we have put more focus on the sensitivity comparison by species in vaccine evaluation.
Point 3: In table 2, add NT50 and additional explanation is required for the asterisk.
Response 3: We have added NT50 and explanation in the asterisk.
579th: NT50: Half maximum neutralization potency
Point 4: In figure 2 and 3, writing more explanation is needed, we think.
Response 4: Thanks for your comments. We have re-checked and supplied more details of figure 2 and 3 as suggested.
Figure 2 (591th): The landscapes of COVID-19 vaccine and five aspects of future development. A: frontier COVID-19 vaccines include inactivated vaccine, viral vector vaccine, etc. B: five aspects of future development: 1, A delicate balance between safety, immunogenicity and durability; 2, Alternation of old and new, common development of multiple lines; 3, Vaccine based on the mucosal immune pathway; 4, Universal coronavirus vaccine designs; 5, Battle with SARS-CoV-2 variants
Figure 3 (678th): Schematic design of PLGA encapsulated oral COVID-19 vaccine: immunogenic antigen are packaged with PLGA in the form of water-oil-water, then modified with M cell receptor targeted peptide, targeting M cell in the small intestine and stimulating mucosal immune response.
Point 5: In 4.5 title, please write more clearly about the prospects.
Response 5: We have written more clearly about the prospects in 4.5.
711th: Preparations should be made to deal with SARS-CoV-2 variants from two aspects. On one hand, closely monitoring the protective ability of existing vaccines on SARS-CoV-2 variants. Once the vaccine loses its protective efficacy on emerging variants, mature technology route should be adopted again. For example, when new variants are added at the feeding end, COVID-19 inactivated vaccine confronting SARS-CoV-2 variants is produced without any changes in production process. On the other hand, the emerging vaccine production technology discussed above may play a role in the rapid response of SARS-CoV-2 variants. Moderna has already initiated development of an updated version of vaccine confronting the SARS-COV-2 mutation. To sum up, mature and flexible vaccine development technology is a key issue in the battle with SARS-CoV-2 variants.
Point 6: In minor, please re-check the number of units in many parts.
Response 6: We have carefully re-check and corrected the number of units in all parts.
Point 7: And there are some mis-spellings, non-grammatical sentences. It will seem to be a more smooth and meaningful sentence after corrections.
Response 7: We apologized for the the error in the manuscript, it is our negligence. We have checked and revised all the contents of the manuscript. If there are still errors, please point out. Thank you very much.
Point 8: Please double-check the position of the exponent used in the table.
Response 8: We have re-check the position of the exponent used in the table.
Point 9: It seems that there are parts that need detail in word selection. For example, RNA and mRNA.
Response 9: We have added the detail in the position of of particular words and paid attention to the selection of words.
Thanks again for your hard work.

Reviewer 3 Report
The submitted manuscript reviewed the current animal models and vaccines of COVID.
Major:
The submission briefly covered two separate topics: COVID animal models and COVID vaccines. There is no relation between these two topics.
There is already a tremendous amount of in-depth reviews about COVID vaccines and clinical trials.
Author Response
Response to Reviewer 3 Comments
Thank you very much for your thorough review of our manuscript. We have carefully addressed each of your comments below in a point-by-point manner.
Point 1: The submission briefly covered two separate topics: COVID animal models and COVID vaccines. There is no relation between these two topics.
Response 1: Thanks for your comments. We didn’t clearly explain the relation between the COVID-19 animal models and vaccines before. We have made revisions in our manuscript.
Firstly, we have emphasized the key role of animal models in vaccine evaluation and their relation before we began to discuss COVID-19 animal models.
67th : In the background of the COVID-19 pandemic, the development of vaccine, antibodies and drugs of COVID-19 were also at high speed. Before clinical trails, the above prophylactic and therapeutic countermeasures are in need of pre-clinical evaluation to ensure their safety and efficacy. COVID-19 animal models are fundamental and essential need in this phase. In the evaluation of vaccines, drugs as well as antibodies, even in the study of the SARS-CoV-2 transmission mechanism, animal models are required to have some common characteristics. For example, the susceptibility to SARS-CoV-2 and the similarity SARS-CoV-2 post-infection response like human beings. Here, we retrospected the basic background and the development paths of COVID-19 animal models. More importantly, we paid close attention to the unique features that animal models needed for vaccine evaluation.
In addition, we have highlighted specific issues in vaccine evaluation when we discussed specific animal models. 117th,198th, 231th,241th.
In summary, we have emphasized animal models for vaccine evaluation whilst discussing the development process of animal models. We will be happy to edit the text further, based on helpful comments from the reviewers.
Point 2: There is already a tremendous amount of in-depth reviews about COVID vaccines and clinical trials.
Response 2: There is indeed a great number of reviews about COVID-19 vaccines from different perspectives. Our review also has its own characteristics.
We focused on popular animal models for vaccine evaluation and sophisticated COVID-19 vaccines. These two parts constitute a sharp weapon against COVID-19. We provided the up-to-date information of them and grasped the detailed commonalities and personalities of current progress and pointed out the future prospects of COVID-19 animal models and vaccines development, aiming to provide readers with the latest progress in COVID-19 animal models and vaccines.
Thanks again for your hard work.
Round 2
Reviewer 1 Report
For the sake of clarity, I enclosed here both the comments raised to the original version (in red) as well as authors' reply (in blue):
Point 1: Unfortunately, this referee feels that the reported data is not suitable for publication as a Review. Indeed, this is a very active line, which is currently ongoing. A large number of groups is developing such animal models, as illustrated in Table 2. Consequently, any review will be quickly out of date.
Response 1: We agree with the reviewer to some extent, but we respectfully hold that our manuscript can be presented in the form of a review. We have made careful revisions to the structure of this manuscript to make it more in line with the criteria of the review. Secondly, COVID-19 animal models and vaccines are indeed developed at high speed, which posed a challenge to keep our manuscript up to date. To overcome this issue, we closely tracked and focused on cutting-edge COVID-19 vaccines and commonly used animal models rather than all related information. To sum up, our manuscript can reflect the background, research status and development trends of our topic, and it meets the stylistic requirements of the review.
Comment 1 to the revised version: Authors' effort to adapt their manuscript to a Review format is welcomed. Although this referee (still) strongly believes the paper betters fit to a perspective contribution in less demanding journal, the additional information might increase the interest of the reported information.
Point 2: I am positive that such information better fits to other format, e.g., a contribution as Perspectives. Please, notice that Vaccines only accepts three types of publications: Articles, Review and Case reports. The submitted manuscript does not fit to any of such formats. I strongly recommend resubmitting to a more suitable journal that allow for such Perspectives manuscript.
Response 2: We respectfully disagree with the reviewer. Our manuscript did make some perspectives. More precisely, they are the prospects of future trends. These prospects are based on our review of the current landscape of COVID-19 animal models and vaccines. So we are positive that the core of our manuscript is to review progress that has been achieved in cutting edge COVID-19 animal models and vaccines, then provide valuable information and prospects. We would prefer it if we submitted our manuscripts to Vaccines. We sincerely believe that our review and prospects on the COVID-19 vaccine and animal models are in line with the requirements of this journal for review articles.
Comment 2 to the revised version: It is not my goal to slow publication down, so that if either Editors and/or my colleagues in the revision process are less skeptic than me about the real impact of such mini-review/perspective paper in the field, I have no further comments/suggestions to add.
Point 3: In addition, all figures must be rendered with a much larger resolution. The current state is quite poor.
Response 3: Thanks for your comments. We have rendered all figures with a larger resolution according to the requirement this journal.
Comment 3 to the revised version: figures are now suitable for publication.
Thanks again for your hard work.
Author Response
Response to Reviewer 1 Comments
Point 1:
Authors' effort to adapt their manuscript to a Review format is welcomed. Although this referee (still) strongly believes the paper betters fit to a perspective contribution in less demanding journal, the additional information might increase the interest of the reported information.
Response 1:
Thank you very much for your hard work and your recognition of our manuscript. Your comments played an important role in the revision of our manuscript. We have carefully considered your comments and made revisions in our manuscript. Besides, we have added some up-to-date information.
We have added the study that robust SARS-CoV-2 infection in nasal turbinates after treatment with systemic neutralizing antibodies in 650th to support the necessity of mucosal immunity.
650th: However, there has been reported that COVID-19 vaccines may not be able to inhibit the infection of the virus in the upper respiratory tract [139].
We have added information about SARS-CoV-2 Delta variant in702nd.
702nd: Among them the Delta variant is spreading all over the world, it’s spike P681R mutation accelerates and enhances S-mediated fusion [164], which may explains the shorter incubation period of Delta variant infection [165]. The control of Delta variant is vital to curb the second wave of the epidemic.
Point 2:
It is not my goal to slow publication down, so that if either Editors and/or my colleagues in the revision process are less skeptic than me about the real impact of such mini-review/perspective paper in the field, I have no further comments/suggestions to add.
Response 2:
Thanks for your comments. We will try our best to render our manuscript a promising review with the help of the editor and reviewers. In addition, we have highlighted our own prospects and characteristics in our manuscript.
Point 3: figures are now suitable for publication.
Response 3:
Thanks for your recognition.
Thanks again for your efforts in our manuscript.
Reviewer 2 Report
I have carefully read the manuscript entitled "COVID-19 animal models and vaccines: current landscape and future prospects" which describes the current status, examples of past use, and future possibilities of using animal models in COVID-19 research.
it seems to be needed more detailed explanations before explaining the whole sentence. For example, we can’t know about other genetically engineered mice in detail like what genes were engineered in the 93rd sentence and about what is the age of the mice from which the fatality rate was derived in the 124th sentence. Also, I recommend writing more detail about what clinical symptoms are mild or moderate: ex) fever, injection area pain, etc in the 299th sentence.
And when considering the unity of the article, there seem to be sentences that can be considered for deletion. Please check again more carefully.
In the 193rd sentence, authors should write about why they are so different to write a meaningful review paper. And because of these two conflicting conclusions, the word ‘successfully’ is not suitable.
In table 1, because there is a difference in quantity, the reliability of the sensitivity comparison by species is lowered.
In table 2, add NT50 and additional explanation is required for the asterisk.
In Figures 2 and 3, writing more explanation is needed.
In the 4.5 titles, please write more clearly about the prospects.
In minor, authors should re-check the number of units in many parts.
And there are mis-spellings, non-grammatical sentences. It will seem to be a more smooth and meaningful sentence after corrections.
Please double-check the position of the exponent used in the table.
It seems that there are parts that need detail in word selection. For example, RNA and mRNA.
And the author should fix the title numbers and font unity.
Author Response
Response to Reviewer 2 Comments
Point 1:
And when considering the unity of the article, there seem to be sentences that can be considered for deletion. Please check again more carefully.
Response 1:
Thanks for your comments. We have checked again carefully and deleted some sentences to make sure the unity of the article.
71st: delete ‘vaccines, drugs as well as antibodies’ and replace it with ‘countermeasures’
77th: delete ‘applied in the evaluation of prophylactic and therapeutic measures’
126th: delete ‘in mice’
208th: delete ‘which have been widely utilized to evaluate the vaccines and therapeutic countermeasures’
291st: delete ‘they represent the promising direction of vaccine development.’
334th: delete ‘Consistent with preclinical results of other inactivated vaccines’
344th: delete ‘The S protein of SARS-CoV-2 has been confirmed to be the most promising antigen for protein subunit vaccine design.’
359th: delete ‘A universal vaccine design to simultaneously overcome COVID-19, MERS and SARS is a promising project.’
475th: delete ‘adopted the previous construction strategy of the MERS vaccine and’
516th: delete ‘suggesting an immune correlate of protection’
528th: delete ‘technology research and development’
599th: delete ‘inactivated vaccines, recombinant protein vaccines, viral vector vaccines as well as nucleic acid vaccines expressing the S as antigen’ and replaced with ‘multiple vaccines based on different design strategies’
655th: delete ‘All the above are the feasibility basis of mucosal immune pathway.’
697th: delete ‘Decreased neutralizing activity against S-G614 pseudovirus of convalescents serum has been reported’
Point 2:
In the 193rd sentence, authors should write about why they are so different to write a meaningful review paper. And because of these two conflicting conclusions, the word ‘successfully’ is not suitable.
Response 2:
Thanks for your comments. We apologized that our previous statement in the 191st sentence was not accurate. In fact, these two studies (References 32, 33) on ferret animal models were not contradictory. Viral RNA shedding in the upper respiratory tract was observed in all ferrets after a high (5×106 pfu or 105.5TCID50) dose of SARS-CoV-2 challenge in the two studies. Low dose (5×102 pfu) challenge was performed in study 2 (References 33), only 1/6 ferrets showed similar signs. We have corrected our statement.
191st: Ferrets recapitulated some typical disease features of COVID-19 observed in humans. Naturally infected ferrets rapidly transmitted SARS-CoV-2 to the entire population via direct or indirect contact [32]. Infected ferrets exhibited elevated body temperatures and virus replication. Virus shedding was confirmed in nasal washes, saliva, urine, and feces while infectious viruses were detected in nasal turbinate, trachea, lungs, and intestine with acute bronchiolitis present in infected lungs. Thus, ferret represented an ideal animal model for virus shedding and transmission. However, mild clinical symptoms and relatively lower virus titers in lungs hindered the fully application of ferret models. Due to the moderate susceptibility to SARS-CoV-2, ferrets were considered a mild clinical disease model of COVID-19. As showed in another previous study [33], Viral RNA shedding in the upper respiratory tract (URT) was observed in all ferrets (6/6) after a high (5×106 pfu) dose of SARS-CoV-2 challenge, while only 1/6 ferrets showed similar signs after low dose (5×102 pfu) challenge. According to the above discussed dose-dependent response to the infection of SARS-CoV-2, the application of ferrets in vaccine evaluation is relatively limited.
Point 3:
And the author should fix the title numbers and font unity.
Response 3:
We have fixed the title numbers and font unity according to your comments.
Thanks again for your efforts in our manuscript.
Reviewer 3 Report
Thank you for the revision.
Author Response
Response to Reviewer 3 Comments
Point 1: Thank you for the revision.
Response 1:
Thank you very much for your hard work in reviewing our manuscript. Your comments are of great significance to the improvement of our manuscript. We have made revisions according to your opinions.
Thanks again for your efforts in our manuscript.
